# The Potential Role of an Aberrant Mucosal Immune Response to SARS-CoV-2 in the Pathogenesis of IgA Nephropathy

**DOI:** 10.3390/pathogens10070881

**Published:** 2021-07-12

**Authors:** Zhao Zhang, Guorong Zhang, Meng Guo, Wanyin Tao, Xingzi Liu, Haiming Wei, Tengchuan Jin, Yuemiao Zhang, Shu Zhu

**Affiliations:** 1Renal Division, Peking University First Hospital, Peking University Institute of Nephrology, Key Laboratory of Renal Disease, Ministry of Health of China, Key Laboratory of Chronic Kidney Disease Prevention and Treatment (Peking University), Ministry of Education, Beijing 100034, China; zhangzhao316@bjmu.edu.cn (Z.Z.); liuxingzi@bjmu.edu.cn (X.L.); 2Department of Digestive Disease, The First Affiliated Hospital of USTC, Division of Life Sciences and Medicine, University of Science and Technology of China, Hefei 230001, China; zhgr@mail.ustc.edu.cn (G.Z.); mengguo@mail.ustc.edu.cn (M.G.); taoustc@ustc.edu.cn (W.T.); 3Hefei National Laboratory for Physical Sciences at Microscale, The CAS Key Laboratory of Innate Immunity and Chronic Disease, School of Basic Medical Sciences, Division of Life Sciences and Medicine, University of Science and Technology of China, Hefei 230027, China; ustcwhm@ustc.edu.cn; 4School of Data Science, University of Science and Technology of China, Hefei 230026, China; 5CAS Centre for Excellence in Cell and Molecular Biology, University of Science and Technology of China, Hefei 230026, China

**Keywords:** SARS-CoV-2, COVID-19, humoral immune response, intestinal dysbiosis and inflammation, IgA nephropathy

## Abstract

The outbreak of severe acute respiratory syndrome coronavirus 2 (SARS-CoV-2) has become a global concern. Immunoglobin A (IgA) contributes to virus neutralization at the early stage of infection. Longitudinal studies are needed to assess whether SARS-CoV-2-specific IgA production persists for a longer time in patients recovered from severe COVID-19 and its lasting symptoms that can have disabling consequences should also be alerted to susceptible hosts. Here, we tracked the anti-SARS-CoV-2 spike protein receptor-binding domain (RBD) antibody levels in a cohort of 88 COVID-19 patients. We found that 52.3% of the patients produced more anti-SARS-CoV-2 RBD IgA than IgG or IgM, and the levels of IgA remained stable during 4–41 days of infection. One of these IgA-dominant COVID-19 patients, concurrently with IgA nephropathy (IgAN), presented with elevated serum creatinine and worse proteinuria during the infection, which continued until seven months post-infection. The serum levels of anti-SARS-CoV-2 RBD and total IgA were higher in this patient than in healthy controls. Changes in the composition of the intestinal microbiota, increased IgA highly coated bacteria, and elevated concentration of the proinflammatory cytokine IL-18 were indicative of potential involvement of intestinal dysbiosis and inflammation to the systemic IgA level and, consequently, the disease progression. Collectively, our work highlighted the potential adverse effect of the mucosal immune response to SARS-CoV-2 infection, and that additional care should be taken with COVID-19 patients presenting with chronic diseases such as IgAN.

## 1. Introduction

SARS-CoV-2 are still spreading and threatening global security. The cellular entry of SARS-CoV-2 is mediated through the angiotensin-converting enzyme 2 (ACE2) receptor [1,2], which is highly expressed in both the lung and the intestine [3,4]. Although pneumonia is the most common symptom in patients with moderate-to-severe illness [5,6,7,8,9,10], 17.6% of COVID-19 patients also develop gastrointestinal symptoms, including diarrhea, anorexia, and nausea [11,12,13,14]. Furthermore, SARS-CoV-2 RNA was reported to be re-detectable in 12 out of 173 patients who had recovered from COVID-19, which was associated with potential SARS-CoV-2 intestinal infection [15,16,17,18]. Besides direct gastrointestinal infection, recent studies have also indicated that SARS-CoV-2 infection leads to intestinal dysbiosis, along with an increase in the numbers of opportunistic pathogens in the intestine [19,20,21].

IgA, which offers humoral protection against microbial pathogens, is the most abundant antibody isotype in the mucosal immune system, including the intestine and lung [22]. SARS-CoV-2-specific IgA was also detected in the tears of COVID-19 patients with different degrees of symptoms and disease stage [23]. Perri et al. found that ocular anti-SARS-CoV-2 IgAs were detected in 10 out of 28 enrolled COVID-19 patients, and that high IgA positivity was observed as early as 2 days after nasopharyngeal/oropharyngeal swabs [23]. Ejemel et al. showed that human anti-SARS-CoV-2 RBD IgA could efficiently neutralize the virus at mucosal surfaces by binding to its spike protein [24]. Sterlin et al. found that IgA dominates the early humoral response against SARS-CoV-2 and contributes to virus neutralization to a greater extent compared with IgG [25]. Besides, specific IgA serum concentrations decreased notably one month after the onset of symptoms [25]. However, as stated that given the time frame covered in their study, further longitudinal studies are needed to assess whether local SARS-CoV-2-specific IgA production persists for a longer time in patients who have recovered from severe COVID-19. Gaebler’s work—a longitudinal analysis of SARS-CoV-2-specific humoral responses in 87 patients—showed that anti-SARS-CoV-2 RBD IgM and IgG antibody titers decrease significantly with IgA being less affected, and that memory B cells with anti-SARS-CoV-2 RBD IgA generating potency persisted for up to six months after infection and exhibited increased neutralization potency [26]. However, lasting mucosal IgA may also promote the pathogenesis of IgA-related diseases, such as IgA-vasculitis with nephritis (Henoch–Schönlein purpura) and Kawasaki disease (KD) [27,28,29,30].

Here, we step further by determining the proportion of patients with IgA-dominant humoral response in defending against SARS-CoV-2 and evaluate the potential role of anti-SARS-CoV-2 RBD IgA in pathology, rather than protection, in a severe COVID-19 patient with concurrent IgAN. To do so, we measured anti-spike protein RBD IgA, IgG, and IgM levels from the serum of 88 COVID-19 patients. We tracked IgA production and renal function longitudinally in a COVID-19 patient with concurrent IgAN, characterized here as a COVID-19 IgAN case, from the serum, urine, and fecal sample. Finally, we characterized the intestinal conditions in this patient. Our results suggest that mucosal immune responses to SARS-CoV-2 infection in the intestine and lung may have worsened renal function and promoted kidney disease progression in this IgAN patient. Extra care should be taken with COVID-19 patients presenting with chronic kidney diseases, such as IgAN.

## 2. Results

### 2.1. More Than Half of the COVID-19 Cohort Produced More Anti-SARS-CoV-2 RBD IgA Than IgG or IgM during SARS-CoV-2 Infection

We first sought to determine the proportion of patients with IgA-dominant humoral response in defending against SARS-CoV-2. Analysis of the anti-SARS-CoV-2 RBD antibodies in the sera of the 88 COVID-19 patients showed that 46 (52.3%) had IgA as the dominant isotype during the infection (Figure 1a). Compared with controls, IgA displayed the highest levels (1,378,338 ± 198,038, *n* = 46 vs. 10,424 ± 747, *n* = 50, *p <* 1 × 10^−4^), followed by IgG (480,603 ± 57,131, *n* = 28, *p =* 0.0009), and IgM (380,694 ± 89,277, *n* = 14, *p =* 0.0082) (Figure 1b). Notably, the IgA level was largely stable within 4 to 41 days of disease onset (Figure 1c).

### 2.2. A Patient with IgA-Dominant COVID-19 and Concurrent IgAN Exhibited Reduced Renal Function during and after Infection

We noticed there was one patient with IgA-dominant COVID-19 who had a history of IgAN and had undergone kidney transplantation 25 months before the infection (the COVID-19 IgAN case). Prednisone (25 mg/day), tacrolimus (5 mg/day), and mycophenolate (0.5 g/day) were administered as post-surgery treatment. The post-surgery urinary protein level was ~0–0.15 g/L and the serum creatinine level was ~160 μmol/L.

On 16 January 2020, the patient was hospitalized due to symptoms including mild fever (37.4 °C), fatigue, and dry cough. No gastrointestinal symptoms such as nausea, vomiting, or diarrhea were observed. Routine virus tests, including influenza A virus, influenza B virus, parainfluenza virus, respiratory syncytial virus, metapneumovirus, coronavirus, rhinovirus, adenovirus, Boca virus, and mycoplasma pneumoniae virus, were all negative. Computed tomography (CT) of the chest showed infectious lesions in both lungs. From 16 to 20 January, the body temperature of this patient fluctuated between 36.9 and 38.0 °C. The white blood cell count (4.2 × 10^9^/L, reference: 3.5~9.5 × 10^9^/L) was within the normal range; the lymphocyte count was low (0.6 × 10^9^/L, reference: 1.5~4.5 × 10^9^/L); and the CD4^+^ T cell count was also low (186 cells/μL, reference: 404~1612 cells/μL). The patient was confirmed to have SARS-CoV-2 infection on January 20 (day 1, d1) by a positive reverse transcription-polymerase chain reaction (RT-PCR) test result for a nasopharyngeal swab. During the infection, the serum creatinine level increased to 197 μmol/L. Worse proteinuria was also reported; however, the 24-h urine protein concentration was not measured due to medical limitations. The patient’s condition deteriorated into respiratory failure and required ventilatory support. Immunosuppression (mycophenolate and tacrolimus) withdrawal was immediately attempted. A combination of anti-inflammatory (methylprednisolone, 40 mg/day), immunity enhancement (human blood gamma globulin, 10 g/day), antimicrobial (moxifloxacin hydrochloride, 400 mg/day and Imipenem and Cilastatin Sodium, 2000 mg/day), antifungal (Posaconazole, 800 mg/day), and antiviral (acyclovir, 250 mg/day/Oseltamivir, 150 mg/day) treatments was attempted immediately. The patient’s condition subsequently improved and stabilized. However, the serum creatinine levels increased to 208 μmol/L at four months, at regular follow-up, and remained high (190–195 μmol/L) even after seven months since the infection was found. The clinical course and renal function characteristics are shown in Table 1 and Figure 2a.

### 2.3. Intestinal Dysbiosis and Inflammation Were Observed in the COVID-19 IgAN Case

We first assessed the anti-SARS-CoV-2 RBD IgA, IgG, and IgM antibody concentrations in the serum, urine, and feces of this COVID-19 IgAN case and compared them with those of healthy controls. The results revealed that all the SARS-CoV-2-specific IgA, IgG, and IgM responses were higher in the serum in this case, with the increase in IgA antibodies being the most significant (Figure 2b). Surprisingly, the IgA antibody levels remained high, even at seven months post-infection (anti-spike protein RBD IgA RLU: 138,475 ± 26,834; anti-spike protein RBD IgM RLU: 18,084 ± 967; anti-spike protein RBD IgG RLU: 36,991 ± 7665) (Figure 2b). However, no increase in anti-SARS-CoV-2 RBD IgA, IgG, or IgM antibodies was observed in the urine or fecal samples of this COVID-19 IgAN case (Appendix A). The anti-SARS-CoV-2 RBD IgA, IgG, and IgM levels in mucosal samples in both the intestine and lung were not measured due to limitations in sample collection.

We then sought to define the intestinal condition of this patient. The total IgA level was markedly higher in both the serum and fecal samples of this patient when compared with that of healthy control (Figure 2c), suggesting that the intestinal dysbiosis accompanied by SARS-CoV-2 infection may have contributed to IgA production. To test this, we performed a 16S rRNA sequencing analysis of fecal samples from the COVID-19 IgAN case and the healthy control. Dysbiosis of gut microbiota was identified, as evidenced by the appearance of the opportunistic pathogens *Pseudomonas* from the family *Pseudomonadaceae* in the fecal sample collected on day 160, which was absent in healthy sample; however, as the patient recovered from the infection, the numbers of these pathogens showed a marked decrease, and were undetectable by day 209. Of note, with recovery, the percentage of *Enterobacteriaceae* significantly decreased and the abundance of *Ruminococcaceae* greatly increased; with these, the overall taxonomic profile in the COVID-19 IgAN case was more similar to that of the healthy control (Figure 2d). Additionally, the concomitant increase in IgA-coated microbiota (Figure 2e) and serum levels of IL-18, a primary mediator of inflammation [31], was also indicative of a more inflamed gut (Figure 2c). Taken together, these results point toward a long-lasting IgA antibody after COVID-19 recovery, along with intestinal dysbiosis, that has potentially contributed to the pathogenesis of IgAN.

## 3. Discussion

In this study, we observed that SARS-CoV-2 infection activated aberrant mucosal humoral responses in both the lung and the intestine, which may have resulted in long-term side effects on the renal function of an IgAN patient who had recovered from COVID-19. Focusing on the humoral response to SARS-CoV-2 infection in a cohort of 88 COVID-19 patients, we found that anti-SARS-CoV-2 RBD IgA was the dominant isotype in the serum of these patients during the infection, with 52.3% showing IgA-dominant COVID-19. Furthermore, among these COVID-19 patients with an IgA-dominant humoral response, we found one who had a history of IgAN who undergone kidney transplantation. An increased serum creatinine level and worse proteinuria were observed during, as well as after, the patient had recovered from the SARS-CoV-2 infection. In addition to severe pneumonia, the patient developed intestinal dysbiosis and exhibited increased production of proinflammatory cytokines in the intestine. Collectively, our work revealed that an aberrant, long-lasting mucosal IgA-mediated response against SARS-CoV-2 may contribute to kidney pathogenesis and IgAN progression (see schematic Figure 3).

IgA production and abnormalities in circulating IgA are the initial events in IgAN pathogenesis [32]. The deposited IgA and the ensuing response of the mesangium are crucial to the development of IgAN [33]. The source of this pathogenic IgA in IgAN is an emerging area of study. Both mesangial and serum IgA are polymeric, and because polymeric IgA is normally produced at mucosal surfaces, this suggests that IgAN is intimately linked with abnormal mucosal immune responses to microorganisms [33]. For example, episodic macroscopic hematuria after upper respiratory infections is common in IgAN patients [34]. Tonsillectomy can improve urinary findings and reduce IgA deposits, thereby exerting a favorable effect on long-term renal survival in some IgAN patients [35,36]. Additionally, IgA is widely distributed in the gut mucosal immune system of IgAN patients, indicating that gut microbiota dysbiosis also plays a role in the pathogenesis of this condition [37,38,39]. Differences in gut microbiota composition have been investigated in patients with IgAN and compared with those of healthy controls [38]. Accordingly, the targeted release of budesonide in the distal ileum was reported to reduce proteinuria and stabilize renal function in IgAN patients [40]. Furthermore, recent studies have reported that the gut microbiota signature of COVID-19 patients was different from that of healthy controls [20]. COVID-19 patients showed significant dysbiosis of the fecal microbiome, which was characterized by an enrichment of opportunistic pathogens and a depletion of beneficial commensals [19]. Numerous experimental and clinical observations suggested that the gut microbiota plays a key role in the pathogenesis of sepsis and acute respiratory distress syndrome, suggesting that SARS-CoV2 might also have an impact on the gut microbiota and vice-versa [41,42,43]. Our results showed that serum and fecal total IgA levels were substantially higher in the COVID-19 IgAN case than in healthy controls, as were the levels of proinflammatory cytokines. This patient also displayed gut microbiota dysbiosis and higher levels of proinflammatory cytokines.

Death is not the only negative consequence of SARS-CoV-2 infection. It has become increasingly clear that many people, even those with mild illness, may develop lasting symptoms that can have disabling consequences. It remains unclear if the virus persists in patients with long-lasting COVID-19, and whether chronic symptoms are due to the direct effects of SARS-CoV-2 infection in multiple organs, or to indirect effects, such as the hyperactivation of the immune system or the development of autoimmunity. Most worrying is the persistence of IgA antibody and memory B cells with IgA-generating potency in patients who have already recovered from COVID-19 [44]. We have yet to fully appreciate their potential for progression to IgAN or even end-stage kidney disease. Thus, special and additional care should be taken with COVID-19 IgAN patients. For example, a complete medical history needs to be obtained for newly diagnosed COVID-19 patients to alert doctors to additional aspects of disease progression. Furthermore, as with the COVID-19 IgAN case reported here, others who have recovered from the disease and who may also be prone to developing IgA-related disease should complete a systematic and regular follow-up to properly monitor disease conditions. Last, but not least, it is important that the people concerned avoid all forms of infections that might result in IgA abnormalities.

We acknowledge that the small patient sample and the lack of evidence from a kidney biopsy in the only COVID-19 IgAN case are the limitations of this study. Nonetheless, the long-term regular follow-up, the strict supervision of the patient’s kidney condition, and our laboratory evidence, all suggest that the high level of mucosal IgA in response to SARS-CoV-2 infection was closely associated with the patient’s reduced renal function, complementing the emerging studies on the aberrant mucosal immune of IgAN. Another limitation includes the lack of samples directly collected from the lung and the intestine, which are unavailable, and other mucosal sites including the eyes and oral cavity. A study found that ocular IgA response against SARS-CoV-2 possibly protects against the progression of infection toward severe symptomatology [23], which may also contribute to the persistent IgA production in this patient.

In conclusion, our study indicated that SARS-CoV-2 infection stimulates aberrant mucosal humoral IgA response in the lung and intestine, and may have contributed to kidney damage and potentially promoted IgAN progression in one patient. Thus, our work highlights the potential adverse effects of the humoral immune response to SARS-CoV-2 infection, and indicates that additional care should be taken with COVID-19 patients with concurrent chronic kidney diseases, such as IgAN.

## 4. Patients and Methods

### 4.1. Patient Cohort

A total of 88 COVID-19 patients, with an average age of 47.35 ± 15.69 years (range 21–91), were enrolled from the First Affiliated Hospital of USTC and the First Affiliated Hospital of Anhui Medical University [45], including one patient with concurrent IgAN who had undergone kidney transplantation at the People’s Hospital of Wuhan University. Specifically, all patients tested positive for SARS-CoV-2 nucleic acid using a real-time fluorescent RT-PCR kit (BGI, Shenzhen, China). Six of the patients were critically ill and were admitted to the intensive care unit; one died of a cerebral hemorrhage after stroke. Seventeen patients had severe COVID-19 and received oxygen supplementation treatment. Fifty-six patients showed moderate illness and nine showed only mild infection. Underlying symptoms were found in thirty-seven (42.0%) of the patients, with hypertension being the most common (18 patients; 20.5%).

A total of 218 serum samples collected during hospitalization and after discharge were tested for SARS-CoV-2 spike (S) protein-specific antibodies. Blood was collected 1, 2, 3, 4, 5, 6, and 7 times for 31 (35.2%), 19 (21.6%), 16 (18.2%), 12 (13.6%), 8 (9.1%), 1 (1.1%), and 1 (1.1%) of the patients, respectively. Fifty archived sera samples collected from healthy donors before October 2019 were used as controls to evaluate the reliability of the measurements. Additionally, urine and fecal samples from the COVID-19 IgAN case were also collected at five months (day 160) and seven months (day 209) after discharge. For the COVID-19 IgAN case, serum, urine, and fecal samples were also collected from five matched healthy controls.

This study was reviewed and approved by the Medical Ethical Committee of the First Affiliated Hospital of USTC (approval number: 2020-XG(H)-014), the First Affiliated Hospital of Anhui Medical University (approval number: Quick-PJ 2020-04-16), and the Peking University First Hospital (approval number: [2020]416).

### 4.2. Measurement of Serum Immunoglobin Levels

Chemiluminescence-based kits to detect SARS-CoV-2-specific IgA, IgM, and IgG antibodies were developed by Kangrun Biotech (Guangzhou, China), in which the receptor-binding domain of the spike protein was coated onto magnetic particles to capture SARS-CoV-2-specific IgA, IgM, and IgG in patient samples [46]. Secondary acridinium-conjugated antibodies were used for detecting IgA, IgM, and IgG. The detected chemiluminescent signals were calculated as relative light units (RLU) (over background signal, the cut-off RLU value for anti-RBD IgA, IgM, and IgG was 32,189, 17,538, and 9971, respectively). These kits have been validated in a large set of serum samples and showed high sensitivities and specificities [45,46]. Serum samples were collected via the centrifugation of whole blood in test tubes at 1000× *g* for 15 min at room temperature. Before testing, a denaturing solution (1% TNBP, 1% Triton X-100) was added to each serum sample. After mixing by inverting, the samples were incubated at 30 °C for 4 h to completely denature any potential viruses. The virus-inactivated serum samples were then diluted 40-fold with dilution buffer and tested at room temperature. The RLU values were then measured using a fully automatic Chemiluminescence Immuno-Analyzer (Kaeser 1000; Kangrun Biotech, Guangzhou, China).

### 4.3. Measurement of Total IgA and IL-18 Concentrations

Total IgA and IL-18 were also detected for the COVID-19 IgAN case. Total serum, urine, and fecal IgA levels were measured using a human IgA-specific ELISA kit (CUSABIO, CSB-E07985h, Wuhan, China) following the manufacturer’s instructions. Serum IL-18 was detected using an IL-18-specific ELISA Kit (Sino Biological, Beijing, China) according to the manufacturer’s instructions.

### 4.4. Flow Cytometric Analysis of Fecal IgA

The fecal samples of the COVID-19 IgAN patient were stored at −80 °C until use. Fecal pellets collected directly from frozen human fecal material were placed in Fast Prep Lysing Matrix D tubes containing ceramic beads (MP Biomedicals, Shanghai, China). A total of 100 mg of fecal material per mL of phosphate-buffered saline (PBS) was incubated on ice for 15 min. The fecal pellets were homogenized by bead beating for 5 s and then centrifuged (50× *g*, 15 min, 4 °C) to remove large particles. The fecal bacteria present in the supernatants were removed (100 µL/sample), washed with 1 mL of PBS containing 1% (*w/v*) bovine serum albumin (BSA; American Bioanalytical, Canton, MA 02021, USA; staining buffer), and centrifuged at 8000× *g* for 5 min at 4 °C). After an additional wash, the bacterial pellets were resuspended in 100 µL of blocking buffer (staining buffer containing 20% normal mouse serum for human samples; Jackson Immuno Research, West Grove, Pennsylvania, PA 19390, USA), incubated for 20 min on ice, and then further incubated with 100 µL of staining buffer containing phycoerythrin (PE)-conjugated anti-human IgA antibodies (1:10; Miltenyi Biotec clone IS11-8E10, Shanghai, China) for 30 min on ice. The samples were then washed three times with 1 mL of staining buffer before flow-cytometric analysis.

### 4.5. 16S rRNA Gene Sequencing

DNA was extracted from fecal pellets using the QIAamp Stool DNA Mini Kit (QIAGEN, Shanghai, China) following the manufacturer’s instructions. The 16S rRNA amplicons were generated using the 515F/806R primer pair recommended by the Earth Microbiome Project (www.earthmicrobiome.org) (accessed on 1 June 2020). The PCR products were quantified, pooled, cleaned using the PCR Cleanup kit (QIAGEN, Shanghai, China), and subsequently sequenced on the Illumina MiSeq platform (V3 kit, PE250 model). The paired-end reads were cleaned and joined to an average depth of 13,442 ± 2526 reads (mean ± SD) and analyzed by Qiime pepline. Raw 16S sequencing data can be found in the NCBI SRA database with the BioProject accession number PRJNA733998.

### 4.6. Statistical Analysis

Data were expressed as means ± SEM. A standard Two-way ANOVA analysis was performed using GraphPad Prism 7. *p*-values ≤ 0.05 were considered significant. The sample size (biological replicates), statistical test used, and the main effects of the statistical analyses for each experiment are detailed in the corresponding figure legends.

## Figures and Tables

**Figure 1 pathogens-10-00881-f001:**
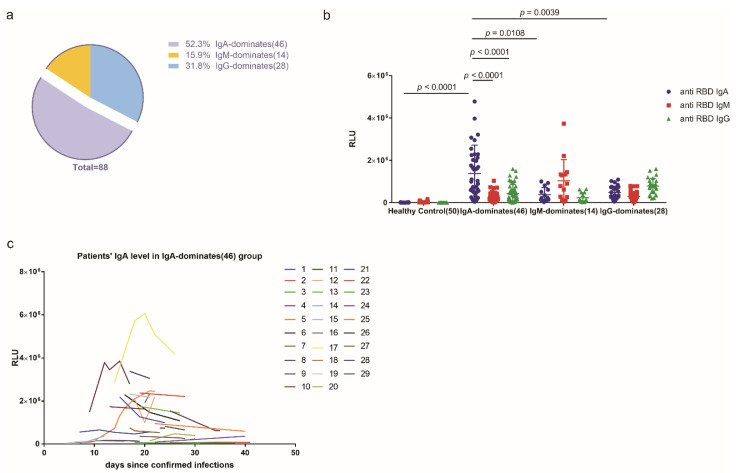
Analysis of the anti-SARS-CoV-2 spike protein RBD antibodies in a cohort of 88 COVID-19 patients during SARS-CoV-2 infection. (**a**). The percentage of patients with IgA-, IgM-, and IgG-dominant COVID-19 in the cohort. (**b**). The relative levels of anti-spike protein RBD IgA, IgM, or IgG in the 88 COVID-19 patients and 50 healthy controls. RLU: relative light unit. The cut-off RLU value (due to background signal) for anti-RBD IgA, IgM, and IgG was 32,189, 17,538, and 9971, respectively. Statistical significance was determined using a Two-way ANOVA analysis. (**c**). The duration of anti-spike protein RBD IgA levels in patients with IgA-dominant COVID-19.

**Figure 2 pathogens-10-00881-f002:**
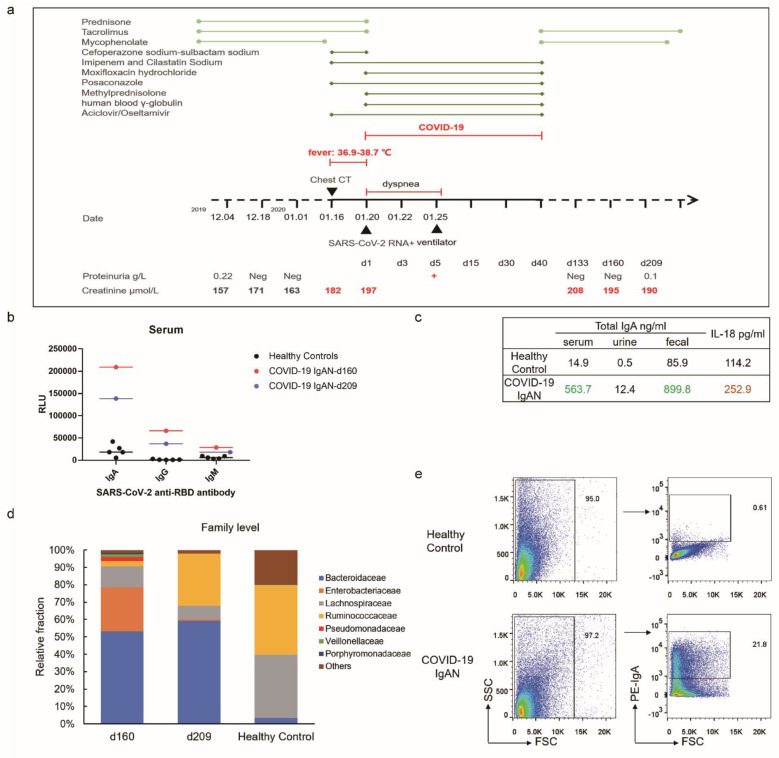
The clinical course of the COVID-19 IgAN case, renal function characteristics and intestinal conditions. (**a**). The clinical course and renal function parameters of the COVID-19 immunoglobulin A (IgA) nephropathy (IgAN) case. (**b**). The serum anti-spike protein RBD IgA, IgM, and IgG at day160 (d160) and day209 (d209) after recovery from COVID-19 in the COVID-19 IgAN case and healthy controls. (**c**). total IgA and IL-18 level in serum, urine and fecal samples from the COVID-19 IgAN case and healthy control. (**d**). 16S rRNA-sequencing of gut microbiota from the COVID-19 IgAN case and healthy control. (**e**). IgA-coated microbiota from the COVID-19 IgAN case and healthy control.

**Figure 3 pathogens-10-00881-f003:**
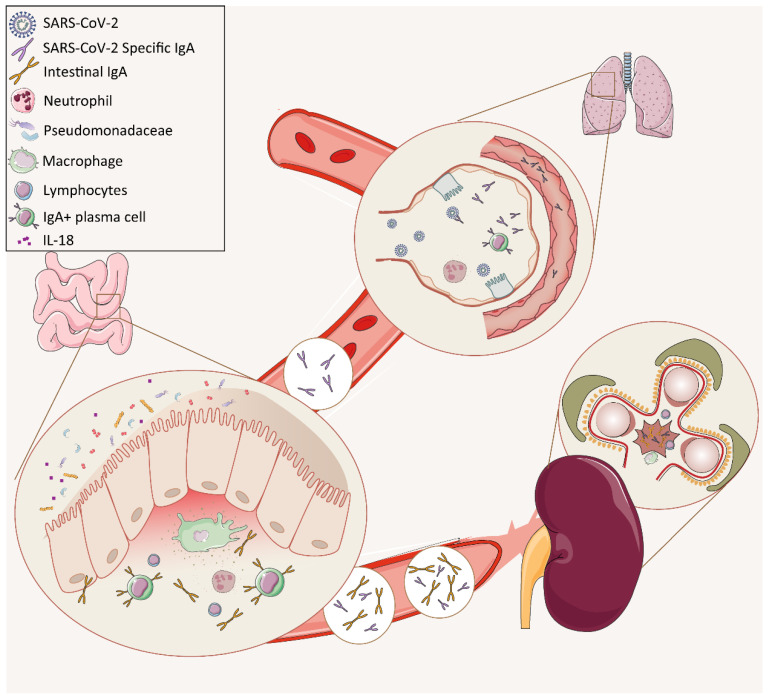
The putative mechanism of the pathogenetic mucosal immunity activated by SARS-CoV-2 in contributing to IgAN progression in the COVID-19 IgAN case. This figure shows the putative mechanism for the pathogenesis of IgA nephropathy during severe acute respiratory syndrome coronavirus 2 (SARS-CoV-2) infection in this specific case. SARS-CoV-2 as an infectious agent activated a primary IgA response in more than 50% patients and the SARS-COV-2 specific IgA could remain stable in the serum for 4–41 days. Meanwhile, SARS-CoV-2 infection was accompanied by intestinal microbiota dysbiosis, which was characterized by an increase in IgA-coated microbiota and several opportunistic pathogens such as *Pseudomonadaceae*. Moreover, the altered microbiota was correlated with the expression of inflammatory factor IL-18. The rising of both SARS-CoV-2 specific IgA and intestinal IgA in serum, as a consequence, contributed to kidney damage and promoted the progression of IgAN.

**Table 1 pathogens-10-00881-t001:** The clinical and laboratory characteristics of the COVID-19 IgAN case.

Clinical Characteristics	COVID-19 IgAN Case
Age, years	39
Sex	Female
Cause of kidney failure	IgAN
Kidney failure vintage, years	14
Signs and symptoms of SARS-CoV-2 infection	
Fever	Yes
Dry cough	Yes
Dyspnea	Yes
Fatigue	Yes
Nausea	No
Vomiting	No
Diarrhea	No
Gross hematuria	No
**Laboratory Characteristics**	
White blood cell count, ×10^9^/L	4.2 (reference: 3.5~9.5)
Neutrophil count, ×10^9^/L	3.2 (reference: 1.8~6.3)
Lymphocyte count, ×10^9^/L	0.6 (reference: 1.1~3.2)
CD4+ cell, cells/μL	186 (reference: 404~1612)

IgAN: immunoglobulin A nephropathy.

## Data Availability

Raw 16S sequencing data can be found in the NCBI SRA database with the BioProject accession number PRJNA733998.

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
