# Peer review of "The Potential Role of an Aberrant Mucosal Immune Response to SARS-CoV-2 in the Pathogenesis of IgA Nephropathy"

_pathogens, 2021, doi:10.3390/pathogens10070881_

Round 1

Reviewer 1 Report

The authors present a work in the form of a Brief report on the potential role of aberrant mucosal immune response in IgA-associated nephropathy.

Although the topic could be interesting, the presented work has several major concerns that need to be addressed before the work can be accepted for publication.

The authors present many data, but it is hard to follow a logical thread in the manuscript, since only some data refer to the cohort of 88 subjects described in the abstract. Actually, most of the paper is focused on one patient only. Actually, the paper is a CASE REPORT rather than a brief report, since the title refers to the data obtained in ONE single patient, and not to a group (although small). In addition, no pathogenic mechanisms are investigated, they are only suggested in the discussion. Also, the authors refer to “mucosal” response but the only data presented regard mostly the serum, and a few data on feces and urine, as antibody levels in the gut and lung were not measured, as the authors declare (lines 414-142).

Assuming that Pathogens may accept case reports, the title itself should be changed to clarify these points: “Aberrant immune response to 2 SARS-CoV-2 detected in a case of IgA nephropathy: clues for pathogenesis”. And manuscript (and the conclusions!) should be rewritten keeping this in mind.

Specific points:

In the introduction it should be cited that mucosal IgA have been detected also at the ocular level (10.3390/biology9110374), likely associated with a better course of the COVID-19 disease. This aspect should be also discussed in the discussion section.

In Figure 1 and in the text between parentheses, how are expressed the results? Titre? Concentration? The samples have been tested in duplicate? Triplicate? How many independent assays?

What is the meaning to measure anti-SARS-CoV-2 antibody levels in uninfected healthy controls that obviously do not have any IgA-IgM-IgG specifically directed against the virus? (Figure 1B)

In Figures 2b and 2c, instead, healthy controls have IgA levels detectable in serum, urine, and feces (together with IL18): how is it possible? Or rather these antibodies are TOTAL antibodies and not specifically directed against SARS-CoV-2? This should be clarified and explained clearly in the text and in the figure legend.

The comparison of the antibody concentration (to support the hypothesis that it is “aberrant” and potentially associated with nephropathy) should be performed between the IgAN case and the other COVID-19 patients (matched for age and gender), and not with healthy controls? What is the meaning of such a comparison with individuals not having a SARS-CoV-2 infection?

Lines 145-146: how can the authors sustain that the intestinal dysbiosis is due to SARS-CoV-2 infection? They observed an association, but they cannot distinguish if the dysbiosis is caused by the virus infection or rather if the virus infection is favored by the intestinal dysbiosis. Second, how could the intestinal dysbiosis may have contributed to IgA production? Please explain

Gut dysbiosis is cited also in the abstract, but there is no detailed description of the microbiome results. They should be described and showed, eventually in a supplementary file. The raw data should be deposited in a data repository cited in the Methods.

In the discussion the authors make several assumptions that are not supported by the presented data and should be rewritten:

line 158-159, the authors did not observe aberrant mucosal humoral response in the lung and intestine, but only an increased (compared to healthy controls) antibody concentration in serum, urine and feces; line168, how can the authors say that the patient “developed intestinal dysbiosis”? do they have samples derived from a previous period to assess whether the intestinal dysbiosis was already present or developed after SARS-CoV-2 infection? Furthermore, was the patient treated with antibiotics? This might have consistently changed the intestinal microbiome causing the observed dysbiosis. This should be clarified and eventually discussed (see also lines 196-197).

Lines 251-252: what are the characteristics of the matched controls? Are they different from the archive sera used for the serum antibody analysis? How were they chosen?

Minor points:

Define the abbreviations “RBD” and “AN” the first time they appear in the text.

Lines 57-58: check the spaces before reference numbers

Line 151: change “decreased” to decrease”

Lines 211-213: The sentence needs a reference

Line 220-222: What is the meaning of this sentence? To avoid infections that potentially cause IgA production or abnormality? How would it be possible for  patient?

Line 223: the work is based on ONE patient, and not on a “small patient sample”; the sentence should be changed and all the work presented as a case report

Line 228: the sentence on “aberrant mucosal immune of IgAN” likely lacks the word “response” and however needs a reference.

Lines 266-269: please define the sensitivity and specificity of the used tests, the background value, the number of duplicate/triplicate samples, the number of independent assays performed

Line 306: please add the public repository where the sequencing raw data are deposited

Author Response

Dear reviewer,

Reviewer 2 Report

The authors present a case with an interesting disease history. It concerns a 39 year old woman, previously diagnosed with IgAN for which she received a transplant. 25 months later she got infected with SARS-CoV-2. The authors hypothesize that her long lasting elevated IgA may contribute to kidney pthogenesis and IgAN progression. Although, as the authors acknowledge as well, the sample size is small and no biopsy was performed, this is an importent message.

Major revisions:

  • statistitcal analysis figure 1a; it is not clear which group is compared to which group since the figure only shows the overvieuw. A figure with detailed datapoints would be helpful.
  • statistical analysis figure 1b; it is not clear which group is compared to which group. If more groups are compared in the same analysis, a student's T test is not fully sufficient and ANOVA should be used.
  • figure 1c. 29 patients are visible while the group consists of 46 patients. Is data of the other patients missing? Which of these is the IgAN case?
  • 16S gut microbiota sequencing in figure 2d is only performed in the IgAN case. To draw conclusions about intestinal dysbiosis, healthy controls or preferably post kidney transplant patients withouth COVID19 infection should be taken into account to rule out influence of medication e.g.

Minor revisions:

  • Lay out figure 1b is not fully visible
  • Mycoplasma pneumoniae is a bacterium and names of micro organisms should be written in italic.
  • Can RLU be calculated into an actual concentration in serum?
  • Exclusion criteria in patient selection are not mentioned

Author Response

Dear reviewer,

Reviewer 3 Report

Zhang et al. operate under the premise that SARS-CoV-2 specific IgA production persists for a longer time in patients recovered from severe COVID-19, which could have deleterious consequences that can seem unrelated to COVID-19. They assessed serum anti-RBD specific antibodies (IgA, M and G) of a cohort of 88 COVID-19 patients. They investigated in more depth one specific patient who had concurrent IgA nephropathy. This patient presented high serum creatinine, proteinuria as well as levels of IgA and RBD-specific IgA higher than healthy controls. For this patient, the authors  also observed changes in the intestinal microbiota, elevated IL-18 and an increase in IgA-coated microbiota.

Overall, the paper is well written, but contain a certain number grammatical and structural errors that can make it confusing at times. The introduction could benefit from  some background information as well as current literature on IgA nephropathy in its normal context as well as in the context of a SARS-CoV-2 infection. The results, in the context of the pandemic are significant and deserve publication. The methods are generally well described.

Major concerns

-The main finding of the paper resides in the description of the clinical case. The information on the cohort of 88 COVID patients have been published by several other groups and represent background information. Thus, this paper should be presented as a case report. 

-The authors should avoid presenting a general mechanism of action based on a single observation. It can still be presented as a tentative explanation of the mechanism of this specific case. 

-The authors do not present mechanistic evidence of the link between COVID-19, the intestinal infection and the promotion of IgAN. The descriptive results do not support the mechanism as many more variables can come into play. For example, ACE2 is expressed in kidney and as such, is predisposed to COVID-19. A recent study showed that almost 44% of patients exhibited renal impairment (Cheng et al. 2020). Thus, although Zhang et al. 's case may suggest a mechanism for sustained IgAN in their specific case, the results do no support the conclusion.

-The authors state in the conclusion that their results show that COVID-19 stimulates aberrant mucosal humoral IgA in the lung and intestine, which has not been demonstrated. The authors collected serum samples, which is representative of the body's net production. The lungs and intestine should represent the majority of this production, theoretically. But it is not what has been shown.

Minor concerns

p2, lines87 and 88: numerical results should be expressed with the units. Also, results should present constant significant numbers (747.3 versus 747).

Figure 1: it might just be the formatting, but significant information on the legend is missing on the right.

Font and figure formatting should uniformized. Example: for figure 1, only "Figure 1" is bold, while in figure 2, the whole title is bold.

is supplementary necessary given that most of the information is already included in the main paper. Given that the focus should mainly be on the case report, all information pertaining to the case should be included in the main article.

page 3 lines 97-98. This sentence is confusing given that it seems like a repetition of what is stated in the next paragraph.

The methods state very clearly that the study was reviewed and approved by two ethics committee. It is not clear however, whether or not the 50 archived sera samples came from patients/donors who signed informed consent.

Author Response

Dear reviewer,

Round 2

Reviewer 1 Report

The authors have answered all the raised concerns and the revised version of the manuscript is considerably improved.

Author Response

We thank the reviewer for the efforts in reviewing our manuscript. 

Reviewer 2 Report

NB

Author Response

We thank the review for the efforts in reviewing our manuscript. 

Reviewer 3 Report

Point 1: The main finding of the paper resides in the description of the clinical case. The
information on the cohort of 88 COVID patients have been published by several other groups
and represent background information. Thus, this paper should be presented as a case report.
Response 1: We thank the reviewer for the suggestion. Our collaborator did publish the same
cohort in another study (Ma et al., Cell Mol Immunol, 2020, 10.1038/s41423-020-0474-z),
however, we re-analyzed the data and present from another angle to emphasize the IgA level
is relative stable from 30-120 days after onset of the disease in each individual, which has not
been shown in the study (Ma et al., Cell Mol Immunol, 2020, 10.1038/s41423-020-0474-z).
So, we would like to keep the data and present the work as brief report.

it can be acceptable to present data that is already in the litterature (predominance of IgA in acute stage of  COVID, not just the 88 cases) if it brings a new angle and if it is required to support the novelty (IgAN case). The authors sustain their position that the results should be presented, which is respectable, although questionable as these data do not bring light on the the mechanism that they want to sell in the discussion and in the conclusion. They do not add value to the case in question. They could still be presented, but with less emphasis on them as it makes the article very confusing and devoid of a logical thread.

-The authors persist on presenting the mechanism, although toned down as the case's mechanism.  I am sensitive to the fact that they highlighted the limitations of the study. The problem is that they do not present any mechanistic investigation. They rather present a series of data that they attempt to link in order to support their conclusion. Even if they did a mechanistic investigation, the authors should avoid drawing mechanistic conclusion based on a single case. 

Author Response

Response to reviewer3-Round 2

Point 1: The main finding of the paper resides in the description of the clinical case. The
information on the cohort of 88 COVID patients have been published by several other groups
and represent background information. Thus, this paper should be presented as a case report.
Response 1: We thank the reviewer for the suggestion. Our collaborator did publish the same
cohort in another study (Ma et al., Cell Mol Immunol, 2020, 10.1038/s41423-020-0474-z),
however, we re-analyzed the data and present from another angle to emphasize the IgA level
is relative stable from 30-120 days after onset of the disease in each individual, which has not
been shown in the study (Ma et al., Cell Mol Immunol, 2020, 10.1038/s41423-020-0474-z).
So, we would like to keep the data and present the work as brief report.

it can be acceptable to present data that is already in the literature (predominance of IgA in acute stage of COVID, not just the 88 cases) if it brings a new angle and if it is required to support the novelty (IgAN case). The authors sustain their position that the results should be presented, which is respectable, although questionable as these data do not bring light on the the mechanism that they want to sell in the discussion and in the conclusion. They do not add value to the case in question. They could still be presented, but with less emphasis on them as it makes the article very confusing and devoid of a logical thread.

-The authors persist on presenting the mechanism, although toned down as the case's mechanism.  I am sensitive to the fact that they highlighted the limitations of the study. The problem is that they do not present any mechanistic investigation. They rather present a series of data that they attempt to link in order to support their conclusion. Even if they did a mechanistic investigation, the authors should avoid drawing mechanistic conclusion based on a single case. 

Response:

We are grateful for the reviewer’s analytical questions. We would like to make every effort to make our contribution and convince the reviewer. Here below are our answers to reviewer’s doubts:

  • The finding from our cohort data provides new information regarding SARS-CoV-2 specific IgA production that over 50% COVID-19 patients produced high amount of IgA antibody and likely associated with disease severity. This is important because the virus is still spreading, and as we have discussed in the text “It remains unclear if the virus persists in patients with long-lasting COVID-19, and whether chronic symptoms are due to the direct effects of SARS-CoV-2 infection in multiple organs, or to indirect effects, such as the hyperactivation of the immune system or the development of autoimmunity. Most worrying is the persistence of IgA antibody and memory B cells with IgA-generating potency in patients who have already recovered from COVID-19” that the infection is damaging and its related symptoms are threatening patients’ lives; Specifically in these IgA-dominant patients, the IgA level remain largely stable individually, which is different from an overall trend as IgA may decline while IgG/IgM will increase to function as neutralizing antibody overtime. These IgA productions may increase the risk of IgA-related disease, such as IgA vasculitis and IgAN. Therefore, timely, proper, and comprehensive clinical observations and treatments are of great importance to treat COVID-19 patients. We step further to find this particular case that there is an IgAN recurrence after COVID-19 infection. This is certainly not an isolated case. Clinicians, especially nephrologist, should be alerted that there are circumstances where COVID-19 infection will lead to acute IgA production and kidney damage in some of the susceptible hosts, and should also be altered the risk of progressive renal failure, which will influence timely decision-making when consider what kind of treatments are to perform.
  • Numerous experimental and clinical observations have reported that the gut microbiota plays a key role in the pathogenesis of SARS-CoV-2 infection. We have done mechanistic studies (with limited materials available from the patient) exploring the intestinal conditions and provided possible mechanism that may have contributed to IgAN progression after COVID-19. There is limited resource where we could find more COVID-19 IgAN patients as the pandemic is well-controlled in China. In countries where the pandemic is ongoing, it is still worth a notification of the complex effects of SARS-CoV-2 even after recovery. We are to highlight the potential harmful effect of mucosal IgA against SARS-CoV-2 and try to provide additional information on how this is leading to kidney damage in susceptible host like patients presenting with chronic diseases like IgAN. Our study is more a complete story highlighting the potential adverse effects of humoral IgA against SARS-CoV-2 in COVID-19 patients with a chronic kidney disease background, it didn’t put much emphasis on the diagnosis or any specific treatment decisions that are normally included in a case report. We would sincerely hope that the reviewer would buy our points.

Round 3

Reviewer 3 Report

Thank you for you detailed response.